# Switching On/Off Amyloid Plaque Formation in Transgenic Animal Models of Alzheimer’s Disease

**DOI:** 10.3390/ijms25010072

**Published:** 2023-12-20

**Authors:** Sergey A. Kozin, Olga I. Kechko, Alexei A. Adzhubei, Alexander A. Makarov, Vladimir A. Mitkevich

**Affiliations:** Engelhardt Institute of Molecular Biology, Russian Academy of Sciences, 119991 Moscow, Russia; olga.kechko@gmail.com (O.I.K.); alexei.adzhubei@gmail.com (A.A.A.); aamakarov@eimb.ru (A.A.M.)

**Keywords:** Alzheimer’s disease, transgenic animal model, amyloidogenesis, amyloid-beta, peptide, isoaspartate, zinc, α4β2 nicotinic acetylcholine receptor, aggregation seeding, anti-amyloid drug

## Abstract

A hallmark of Alzheimer’s disease (AD) are the proteinaceous aggregates formed by the amyloid-beta peptide (Aβ) that is deposited inside the brain as amyloid plaques. The accumulation of aggregated Aβ may initiate or enhance pathologic processes in AD. According to the amyloid hypothesis, any agent that has the capability to inhibit Aβ aggregation and/or destroy amyloid plaques represents a potential disease-modifying drug. In 2023, a humanized IgG1 monoclonal antibody (lecanemab) against the Aβ-soluble protofibrils was approved by the US FDA for AD therapy, thus providing compelling support to the amyloid hypothesis. To acquire a deeper insight on the in vivo Aβ aggregation, various animal models, including aged herbivores and carnivores, non-human primates, transgenic rodents, fish and worms were widely exploited. This review is based on the recent data obtained using transgenic animal AD models and presents experimental verification of the critical role in Aβ aggregation seeding of the interactions between zinc ions, Aβ with the isomerized Asp7 (isoD7-Aβ) and the α4β2 nicotinic acetylcholine receptor.

## 1. Introduction

A most common type of dementia in older adults, Alzheimer’s disease (AD), is a severe neurodegenerative pathology manifested by progressive cognitive decline, such as memory loss and suppressed logical thinking [1]. Currently, AD is defined by the following neuropathological profile: (1) deposition of amyloid-beta peptide (Aβ) aggregates in the form of diffuse and amyloid (neuritic, senile) plaques [2] and (2) the presence of intraneuronal neurofibrillary tangles and neuropil filaments (in dystrophic neurites) comprised of aggregated hyperphosphorylated tau protein [3,4]. These pathomorphological features are observed in certain areas of the brain [5,6], but are particularly common in the hippocampus, an area of the brain critical for learning and memory, where amyloid plaques, neurofibrillary tangles and neuropil threads appear at the earliest stages of AD [7,8]. In addition to classical neuropathological features, AD is characterized by systemic abnormalities and disorders of brain metabolism that appear at the molecular and biochemical levels and include cholinergic failure [9], neuroinflammation, activation of apoptosis, mitochondrial dysfunction, metabolic disorders and chronic oxidative stress [10].

Hereditary AD variants, which are characterized by the early onset and more rapid progression, account for less than 1% of all cases of this pathology and are associated with mutations [11,12] that lead to an excess of physiologically normal Aβ levels. Sporadic AD variants, which are characterized by later onset and a relatively slow progression, affect over 95% of patients and, similar to the inherited variants, are closely connected with the abnormal aggregation of endogenous Aβ [2]. Compared with the patients with late-onset Alzheimer’s disease, patients with familial AD variants have more amyloid plaques and more developed cerebral amyloid angiopathy [13].

Normal endogenous Aβ is a small polypeptide molecule of 39–43 amino acid residues [14]. Aβ is produced by proteolysis of the amyloid precursor protein (APP) [14] and is present in both the brain tissue and peripheral organs [15]. The physiological roles of Aβ may include regulation of the synaptic function, protection against infection, repair of the damaged areas in the blood–brain barrier, and compensatory role for the effects of injury [16]. In the process of the AD pathogenesis, soluble dimers and oligomers of Aβ appear in biological fluids of the body. These Aβ species, when they bind to cellular receptors, cause the dysfunction and degeneration of synapses [17]. Presumably in the later stages of AD, Aβ oligomers stay in dynamic equilibrium with aggregated Aβ molecules of the amyloid plaques [18]. The most commonly occurring sequence in amyloid plaques Aβ42 contains 42 amino acid residues [19]. In addition to Aβ and its chemically modified isoforms [20], amyloid plaques include many other components, such as proteoglycans, carbohydrate-binding proteins of the innate immune system, nucleic acids, biometal ions, lipids, and transport proteins [21]. It is believed that such components can seriously affect the processes of Aβ aggregation in AD pathogenesis [22].

Analysis of the morphology of amyloid fibrils isolated from the brain tissue of patients diagnosed with AD showed that, despite the polymorphism of the fibrils, they all consist of protofilaments with a similar structure [23]. The spatial structure of soluble Aβ monomers and oligomers cannot be obtained experimentally under the physiologically relevant conditions due to the spontaneous aggregation of Aβ at the concentrations required for modern physicochemical methods [24]. It is generally accepted that the conformational transformation and aggregation of monomeric Aβ molecules occurs via the nucleation mechanism [25,26,27]. According to this mechanism, a slow and thermodynamically unfavorable nucleation phase is followed by the fast polymerization phase [28]. In the nucleation phase, the stage that determines the integral rate of the entire Aβ aggregation is the formation of a stable nucleus of the polymerized protein. The nucleus must necessarily contain Aβ in an oligomeric state, albeit additional molecular agents, which along with Aβ are present in amyloid plaques, seem to play highly important role in the appearance of the seed of pathological Aβ aggregation [29].

The primary current hypotheses on the etiology of AD are (reviewed in [30]) (1) “amyloid cascade” [31]; (2) “protein aging” [32]; (3) “cholinergic deficit” [9]; (4) “zinc dyshomeostasis” [33]; and (5) “inflammatory” [34]. Yet, the prevailing experimental evidence on the pathological physiology of AD supports the amyloid cascade hypothesis for AD pathogenesis [31,35]. This hypothesis postulates that the accumulation of Aβ aggregates in the brain (cerebral amyloidogenesis) triggers a signaling cascade that causes pathological transformation of the tau protein, neuroinflammation, and neurodegeneration. Consequently, the appearance and spread of the extracellular Aβ aggregates (amyloid plaques) in brain tissue is one of the main pathological signs for both sporadic and hereditary variants of AD and, possibly, constitutes the primary pathogenetic process of AD [36].

The recently introduced Amyloid Cascade Hypothesis 2.0 (ACH2.0) interpretation of AD states that extracellular Aβ in general and Aβ plaques in particular are largely benign and, possibly, even physiologically protective, and that AD is actually driven by intraneuronal Aβ (iAβ) [37,38,39,40]. The causative role of iAβ is the central postulate of the ACH2.0, which envisions AD as a two-stage disease. In the first pre-symptomatic stage, AβPP-derived Aβ accumulates to critical levels in a decades-long process that induces the activation of the second, devastating symptomatic AD stage that is anchored and driven by an agent which is independent of AβPP and sustains and perpetuates its own production [37,38]. The crucial role of the AβPP proteolytic/secretory pathway in only the first, pre-symptomatic stage of AD explains why the drugs targeting extracellular AβPP-derived Aβ or its production by the AβPP proteolysis did not and indeed could not have any effect on symptomatic AD patients. The progression of the disease is driven at this stage by iAβ produced independently of AβPP and the drugs fail despite effectively fulfilling their mechanistic purpose. By the same logic, the overall success of the same drugs in the animal models suggests that no second AD stage occurs there, consistent with the inability to obtain full spectrum of AD pathology in those experimental systems [40]. However, the formation of amyloid plaques seems to occur through the same molecular mechanism in both humans and the animal models of Alzheimer’s disease.

From the time of the formulation of the amyloid hypothesis of AD in the early 1990s [31] and for many years, all drugs aimed at the prevention of Aβ aggregation failed in clinical trials [41]. As a result, until 2015 most pharmaceutical companies significantly limited further development of the anti-amyloid drugs aimed at the treatment of AD, and the amyloid hypothesis was discredited among a significant part of researchers [42,43]. However, full approval by the US FDA of the anti-Aβ antibody lecanemab (marketed as Leqembi) [44] in July 2023 marked a turning point in Alzheimer’s disease research and demonstrated the clinical benefits of anti-amyloid therapy [45,46,47].

This review addresses the advances in understanding the molecular mechanism of the formation of amyloid plaque in the nervous system of model animals and offers an analysis of the possibilities of using the structural determinants of amyloid-beta as drug targets for anti-amyloid therapy for Alzheimer’s disease.

## 2. Animal Models of Alzheimer’s Disease

The vast majority of AD cases have late onset, but their clinical and histopathological features are common with the early autosomal-dominant AD variants, which are the main target of animal modeling [48,49]. Autosomal-dominant variants of AD are associated with mutations in the genes that encode proteins involved in the generation of Aβ, including the *APP* gene, encoding amyloid-beta precursor protein (APP), as well as the *PSEN1* and *PSEN2* genes, encoding presenilin-1 (PS1) and presenilin-2 (PS2). APP is a transmembrane protein with the purported but as yet unclear roles in several aspects of the neuronal homeostasis. Most APP mutations cluster near the sites typically cleaved by proteases: α-, β-, and γ-secretases. PS1 and PS2 are components of the gamma-secretase complex responsible for the proteolytic cleavage of APP. AD-associated mutations in the genes APP, PSEN1, and PSEN2 promote Aβ formation by favoring proteolytic processing of APP via the β- or γ-secretase pathway rather than via the non-amyloidogenic α-secretase pathway. In the amyloidogenic pathway, APP cleavage is initiated by β-secretase. Subsequent intramembrane cleavage by γ-secretase results in the formation of 40- and 42-amino acid amyloid peptides (Aβ40 and Aβ42, respectively) The longer forms (Aβ42 or longer) display a greater tendency to self-aggregation [50]. These proteolytic processes and subsequent modifications result in the appearance of Aβ in the form of dimers, tetramers, oligomers, protofibrils, and amyloid fibrils [51].

The recognized in vivo AD models can be divided into spontaneous, interventional, and genetically modified [52,53]. Aβ accumulation and tau hyperphosphorylation can occur spontaneously in non-human primates (NHPs). For example, baboons show only a formation of neurofibrillary tangles, while macaques demonstrate amyloid deposition without evidence of tau pathology. However, these NHPs have a long lifespan, and spontaneous AD-like symptoms and pathological changes are usually observed only in the older adults. Consequently, although the spontaneous AD models using NHPs have research value, various factors such as high maintenance costs, low reproductive potential, handling problems, and risk of zoonotic transmission limit the use of these models. In the interventional models, chemicals are used to induce symptoms and pathological changes similar to those observed in the AD pathogenesis. In particular, in the interventional AD models, various chemicals are utilized to induce neuroinflammation [34].

The earliest animal AD models were created by disrupting the cholinergic system of the basal forebrain in various mammalian species using surgical techniques [54], neurotoxins, immunotoxins, and pharmacological methods. The target species included mice and rats, rabbits and monkeys [55]. These models reproduced some symptoms of AD, such as memory impairment, and were useful for testing the effectiveness of cholinesterase inhibitors, which may furnish some symptomatic relief at the early stages of AD. In these models, of course, neither plaques nor neurofibrillary tangles (NFT) were formed, nor did they reflect the development of the complex biochemical and cellular changes in AD. As a result, such models gradually lost their relevance in the AD research.

The spontaneous animal AD models, unlike transgenic AD models, do not include mutations associated with the hereditary AD variants, but are based on developing the pathology, which is accompanied by a decline in cognitive functions in old age. Therefore, spontaneous models can be used to evaluate the therapeutic approaches and to diagnose sporadic AD, which is the predominant form of AD. Amyloid plaques are found in the brains of the aged animals [56,57], indicating that this pathology is a common accompaniment of aging not only in humans, but also among other species. Neuritic plaques and cerebrovascular amyloid deposits were found in the aged monkeys, dogs and polar bears but were rarely found in aged rodents [58]. Larger animals naturally develop amyloid plaques and/or NFT as they age, and these features may be normal in larger animals. They were found in many species of primates, as well as in a number of large herbivores and carnivores. This propensity to form plaques appears to be due, at least in part, to the sequence conservation of the amyloid-beta peptide in most mammals [59] with the exception of rodents (mice and rats) [60]. In contrast to the simple demonstration of amyloid deposits, though the tau-associated tangle pathology was demonstrated in some species [61], it is much less common there than in the human brain. Thus, spontaneous animal AD models could potentially be useful to bridge the gap between the promising rodent studies and failed human clinical trials [62].

Evidently, there is no ideal animal model of Alzheimer’s disease, and any given model only reflects certain aspects of the condition of a patient with AD. Mice species are most commonly used due to their ease of breeding and genetic manipulation, as well as the relatively low cost of maintenance [48]. However, wild-type mice do not develop tau or Aβ pathology, possibly because the rat/mouse amyloid-beta differs from human amyloid-beta and that of most other vertebrates by three amino acid substitutions in the metal-binding domain 1–16, which significantly affects Aβ structural and functional properties [63]. AD models where the genetically altered (transgenic) rodents are utilized have provided fundamental insights into the molecular mechanisms of inherited variants of Alzheimer’s disease [64]. Additional vertebrate species used for modeling employing transgenic approaches include the rat (better suited for behavioral testing than mice), the sea lamprey (which has giant driven neurons), and the zebrafish with its transparent larvae that make neurons easier to visualize. Invertebrate species that are also used include the fruit fly, and the roundworm (both of which are suitable for screening of the developed drug’s effectiveness). Neurodegeneration has been successfully modeled in pigs and sheep, with the features of AD pathology modeled in transgenic pigs [65].

Currently, there are approximately two hundred spontaneous and genetically modified AD models [66], which are based on utilizing various animals, including transgenic mammals (which overexpress human genes involved in the formation of amyloid plaques and neurofibrillary tangles [67]), as well as transgenic flies [68], worms [69], and fish [70]. Mice is used in the majority of the animal AD models, including transgenic, knockout, and injection models [71]. Wild-type mice do not form senile plaques or neurofibrillary tangles even in old age and therefore cannot be used as model animals for AD. Based on the genetic evidence that nearly all mutations associated with inherited AD variants are associated with alterations in Aβ production or aggregation, advances in genetic engineering technologies have enabled the development of mouse models using the *APP* and *PSEN1* genes. Such models are characterized in the first place by the presence of fibrillar and diffuse amyloid plaques. The most popular models include mice of the following lines: PDAPP, Tg2576, APP23, J20, TgCRND8, PS2APP, APPswe/PSEN1dE9 (APP/PS1), Tg-ArcSwe, 5xFAD, A7, and AppNL-G-F. Mice where the neuronal dysfunctions are developed in association with the hyperphosphorylated forms of tau, as a result of transgenes with mutations in MAPT, include the following strains: JNPL3, PS19, and rTg4510. The 3xTg mice were engineered by co-injection of the two genes carrying the linked mutations [APP with the Swedish mutation (KM670/671NL) and MAPT with the P301L mutation]. As a result, these mice develop progressive neuropathology, with intracellular and extracellular Aβ deposits and the aggregates of phosphorylated tau characterized by conformational changes.

## 3. Molecular Factors Affecting Amyloid Plaque Formation in Transgenic Animal Models of Alzheimer’s Disease

### 3.1. Zinc

A number of observations indicate that Aβ interactions with zinc ions are involved in the pathogenesis of AD [72]: (1) amyloid plaques contain abnormally high amounts of zinc ions [73]; (2) zinc binds to Aβ and causes its rapid aggregation [74], possibly by modulating the conformational transformation of Aβ and the population shifts in the equilibrium of Aβ polymorphic states [75]; (3) in postmortem brain tissue samples from the patients diagnosed with AD, areas of increased zinc concentration coincide with the sites of amyloid plaque formation [76]; (4) brain regions most affected by the AD pathology contain dense innervation by zinc-containing axons, while brain regions less affected by pathology contain negligible amounts of zinc-containing terminals [77]. Zinc is the second most abundant micronutrient in the human brain (after iron) [78] and serves as an essential structural and catalytic component for more than 10% of the proteins encoded in the human genome [79]. Under physiological conditions, zinc is involved in modulating synaptic plasticity, can regulate neurogenesis, neuron migration and differentiation, and plays a role in neurotransmission [80]; it also has neuroprotective properties and can protect against oxidative stress [81,82]. Hence, the physiological functions of zinc and Aβ largely overlap, and during the pathogenesis of AD, zinc and Aβ constitute the main components of the amyloid plaques and are in direct contact with each other.

The concentration of zinc ions in the amyloid plaques reaches 1 mM [73]. It is believed that the source of zinc that accumulates in the plaques is glutamatergic synapses [83]. Free zinc ions are present in high concentrations (up to 0.3 mM) in the presynaptic vesicles of glutamatergic neurons of the limbic system and cerebral cortex [84]. Upon excitation, zinc ions at a concentration of 10–100 μM enter the synaptic cleft of neurons along with glutamate molecules, where they interact with NMDA and AMPA receptors, modulating their function [85]. To confirm the important role of zinc located in the synaptic cleft, the formation of amyloid plaques was analyzed in transgenic mice with the zinc transporter ZnT3 protein inactivated, and it was found that, in such mice, plaque formation was sharply reduced [83].

The ability of Aβ molecules to bind zinc ions in vitro was first demonstrated in 1994 using Aβ40 as an example [74]. For a long time, it was not possible to establish the mechanism of interaction of Aβ with zinc ions owing to the rapid zinc-induced aggregation of both the most common, in the organism, full-length variants of Aβ, i.e., Aβ40 and Aβ42, and of the truncated at the C- terminus Aβ isoforms, particularly Aβ28 [74,86,87,88,89,90]. However, the N-terminal fragment 1–16 (Aβ16), which is also present in humans as an independent peptide [91], is an autonomous metal-binding domain of Aβ that is stable in vitro [92]. The structure of this domain was determined in the free state and in complex with a zinc ion under physiologically relevant conditions [93]. Then, the molecular mechanism of interaction between zinc ions and Aβ [94] has been established: (1) firstly, zinc ion is chelated by side groups of the amino acid residues Glu11, His13 and His14; (2) followed by folding of the N-terminal region 1–8 of Aβ16, and further by the formation of an additional coordination bond between the zinc ion and the side group of the His6 amino acid residue; this results in the appearance of a well-ordered compact structure of the entire 1–16 domain.

Importantly, the 11-EVHH-14 region of Aβ acts not only as the primary zinc ion recognition site, but also controls the processes of zinc-induced Aβ dimerization [95,96] and oligomerization [97]. The 11-EVHH-14 region has a relatively rigid backbone conformation in soluble Aβ monomers [93,98] and zinc-bound dimers [97]. This site corresponds to the β-strand β2 from the N-terminal arch of the Aβ amyloid fibrils purified from Alzheimer’s brain tissue and is solvent-exposed and accessible for interactions with the external molecules [23]. The secondary structure of the 11-EVHH-14 region is a left-handed polyproline II-type helix [99], which has an increased propensity to participate in the protein–protein interactions [100]. On the strength of these data, the 11-EVHH-14 site represents a structural determinant of Aβ.

### 3.2. α4β2 Nicotinic Acetylcholine Receptor

It was established using bioinformatic approaches that the extracellular N-terminal domain of the α4 subunit of α4β2-nAChR includes the fragment 35-His-Ala-Glu-Glu-38 (35-HAEE-38), ion-complementary to the 11-EVHH-14 region of Aβ. The fragment 35-HAEE-38 (conservative for humans, mice and chicken) forms the interaction interface of α4β2-nAChR with Aβ [101]. It was shown by molecular modeling that this interaction is stabilized by ion bridges between the complementary charged side groups of the H/E and E/H amino acid pairs. Thus, Aβ can interact with α4β2-nAChR via the 11-EVHH-14: 35-HAEE-38 interface. Tetrapeptide Ac-HAEE-NH_2_, which is the synthetic analog of the receptor side of this interface, was proven to bind Aβ and to efficiently repair the Aβ-dependent loss of cholinergic function in α4β2-nAChR-transfected oocytes [101]. The data suggest that Aβ:α4β2-nAChR interaction is mediated by the charge complementary interface (Aβ) 11-EVHH-14:35-HAEE-38 (α4), and that the interaction may be involved in Aβ aggregation seeding. So, one can rationally suggest that the absence of α4β2-nAChR would suppress amyloid plaque formation.

Indeed, study [102] reports a number of protective effects caused by the loss of α4*-nAChRs on the neuropathological alterations that develop over time in Tg2576 (APPswe) mice, a widely studied mouse model of AD that expresses a human APP transgene carrying the amyloidogenic Swedish mutation. The principal effect of α4KO was to markedly reduce the load of Aβ plaques in neocortical areas. The plaques were reduced in number, but their size distribution was unchanged, suggesting that fewer plaques are initially formed, but once they are seeded, their maturation is not altered by the loss of α4*-nAChRs. This evidence supports the hypothesis that α4*-nAChRs are directly involved in the seeding of the plaques without affecting plaque maturation and the specific Aβ isoform processing [102]. 

### 3.3. Amyloid-β with the Isomerized Asp7 (isoD7-Aβ)

One of the main signs of the pathogenesis of AD is cerebral amyloidogenesis (CA), the process of forming amyloid plaques in the patient’s brain [3,4]. The ability of Aβ aggregates isolated postmortem from the brains of patients diagnosed with AD to induce the development of cerebral amyloidosis was first shown in monkeys that were intracerebrally injected with the corresponding autopsy material [103,104]. Further, in a series of studies on transgenic rodent models of AD [105,106,107,108], it was established that the molecular agent that causes the formation of amyloid plaques in the brain tissue is the conformationally or chemically modified version of Aβ [105,106,107,108,109,110,111,112]. It was also shown that, upon induction of CA, amyloid plaques form in model animals within 24 h, followed in 1–2 days by the activation of microglial cells and their migration to the plaques, which is accompanied by local changes in the axons and dendrites of nearby neurons [113]. The spread of amyloid plaques throughout the brain occurs through the extracellular pathway [114]. As a result of incubation in hippocampal slice culture, synthetic Aβ is converted into a pathogenic agent that causes CA in vivo via a seeding mechanism (in the presence of small amounts of the brain homogenate from the AD patients) [115].

The seeding mechanism of CA initiation in AD involves an important change in the native structure of endogenous Aβ due to the interaction with pathogenic Aβ molecules [29] present in amyloid plaques. This mechanism is corroborated by the fact that small amounts of the pathologically altered Aβ molecules contained in the material obtained from patients diagnosed with AD cause the transition of native Aβ molecules to pathological state [116,117].

It was established that injections of homogenized brain preparations from patients diagnosed with AD lead to a sharply accelerated CA in model animals [29]. Of note, amyloid-β with isomerized Asp7 (isoD7-Aβ) accounts for over 50% of all Aβ molecules in the amyloid plaques [118] and, accordingly, such homogenates contain significant amounts of isoD7-Aβ. IsoD7-Aβ is formed spontaneously due to “protein aging” [32] of both the circulating and aggregated Aβ molecules [14]. It was shown that the accumulation of isoD7-Aβ in brain tissue is a common process in elderly subjects, but patients diagnosed with AD have significantly higher levels of isoD7-Aβ [119]. IsoD7-Aβ42 was also shown to be toxic to neuronal cells [120,121]. IsoD7-Aβ cytotoxicity is coupled to protein phosphorylation [122] and to the induction of oxidative stress, actin polymerization, and stress fiber formation [123]. Molecular mechanism of the in vitro formation of zinc-induced oligomers of natural Aβ isoforms [97] imply that isoD7-Aβ can act as a nucleation seed of aggregation of the endogenous native Aβ molecules in the presence of high concentrations of free zinc in vivo, e.g., in the clefts of cholinergic neurons, where the level of zinc ions reaches 300 µM [85,124].

Indeed, in study [125], it was shown that, unlike Aβ42, isoD7-Aβ42 when administered intravenously to males of transgenic mouse strain B6C3-Tg(APPswe, PSEN1-dE9)85Dbo/j (APP/PS1) caused a sharp acceleration in the development of CA. The experimental groups included male animals that were raised in specific pathogen-free conditions and received monthly intravenous injections of isoD7-Aβ42 (at doses of 100 μg) starting at 2 months of age. Then, the amount of fibrillar amyloid plaques in the hippocampus, cortex and thalamic nuclei was determined in the brain samples from 4- and 10-month-old animals. In contrast to the 4-month cohorts of mice, in which the number of amyloid plaques in the hippocampus and cerebral cortex was approximately 9 and 4 times higher in the animals injected with isoD7-Aβ42 than in the intact control animals, the 10-month cohorts showed for these areas of the brain a smaller difference of about 3 and 1.5 times, respectively. In the thalamic nuclei, the difference between the injected and intact 10-month-old transgenic mice was again very high, by 5.2 times, while the 4-month-old transgenic mice did not have amyloid plaques in this area of the brain. Unlike transgenic mice, intravenous injection of isoD7-Aβ42 to the wild-type mice did not cause the formation of amyloid plaques in the animals, regardless of their age. Thus, it was shown that isoD7-Aβ42 plays the role as a nucleus for the aggregation of endogenous Aβ molecules during the development of CA in a transgenic model of AD [125].

It was later found that intracerebral administration of the metal-binding domain 1-16 of isoD7-Aβ to the transgenic 5xFAD mice substantially enhanced CA [126]. At the age of 3 months, two groups of male 5xFAD transgenic mice were given a single intracerebral injection of isoD7-Aβ16 or Aβ16 solution using the stereotaxic procedure. Brains were removed 1 month after the stereotaxis procedure, and the 8-μm-thick sagittal sections of the brain were analyzed histochemically using Congo red staining. The mean number of congophilic amyloid plaques in the group of transgenic mice treated with the isoD7-Aβ16 was 19.9 ± 2.0, while the mean number of the congophilic amyloid plaques in the group of animals treated with Aβ16, was 12.9 ± 2.1. Hence, isoD7-Aβ16 was identified as a minimal seed for the zinc-dependent aggregation of endogenous Aβ molecules [126].

It was demonstrated that passive immunization of the transgenic 5xFAD mice by the specific monoclonal antibodies recognizing isoD7-Aβ resulted in a significant reduction of isoD7-Aβ and total Aβ in the brain [127]. Amelioration of cognitive impairment was demonstrated by the Morris water maze, elevated plus maze, pole, and contextual fear conditioning tests. Consequently, the antibody-mediated targeting of isoD7-Aβ peptides leads to the attenuation of the AD-like amyloid pathology.

## 4. Molecular Tools Switching On/Off the Aggregation of Endogenous Aβ Molecules in a Transgenic Model of Alzheimer’s Disease

### 4.1. Neither Zinc Nor isoD7-Aβ, but a Mixture of Them Triggers Amyloidogenesis

Transgenic nematodes *C. elegans* that overexpress human Aβ are a popular animal model in the studies of cerebral amyloidogenesis in AD [128,129,130,131]. As these nematodes age, fibrillar amyloid plaques appear in various tissues of the body, and various functional abnormalities are observed. However, the average life span expectancy of animals remains almost unchanged, which indicates only a limited effect of the constitutive amyloid plaques on the main functions of nematodes. One explanation for the relative harmlessness of the amyloidogenesis for nematodes may be the insufficient amounts of molecular components that are present in the human amyloid plaques, namely, zinc ions and isoD7-Aβ. Under the normal nematode life conditions, the concentration of zinc ions in their body cannot reach peak values [132,133]. Also, the occurrence of isoD7-Aβ in nematodes is unlikely, since spontaneous isomerization of Asp7 within Aβ requires substantial time [134], exceeding the life span of these animals.

The study [135] analyzed changes in the three main experimental characteristics associated with the development of amyloidogenesis in transgenic nematodes *Caenorhabditis elegance* CL2120 (integral volume of amyloid plaques in tissues; degree of muscle paralysis; and life expectancy) when several molecular agents (Aβ, isoD7-Aβ, and zinc ions), which are present in amyloid plaques of the patients diagnosed with AD were added to a nutrient medium. Any single agent by itself had no effect on the animals. A binary mixture of Aβ and zinc ions was also harmless to nematodes. However, the concurrent addition of zinc ions and isoD7-Aβ led to a sharp increase in the amyloid load, an increase in the degree of paralysis and a shortening of the lifespan of animals. Therefore, it was demonstrated in vivo that the isoD7-Aβ/zinc complex has a determinative role in triggering the chain process of zinc-induced aggregation of endogenous Aβ molecules, associated with the severe functional consequences for the life of experimental animals.

### 4.2. In Transgenic Nematodes, nAChR α4-Derived Peptide HAEE Neutralizes the Aggregation Seeding Effect of the Zinc and isoD7-Aβ Mixture

On the basis of the molecular mechanism of zinc-dependent oligomerization of Aβ isoforms [97], it can be stipulated that molecular agents that can, together with the 11-EVHH-14 sites of Aβ chelate the common zinc ion, will inhibit zinc-dependent oligomerization of Aβ. Indeed, it was shown [135] that the presence of the tetrapeptide Acetyl-HAEE-NH_2_ (HAEE) completely neutralizes the negative effects of the mixture of zinc ions and isoD7-Aβ on the quality of life of nematodes *Caenorhabditis elegance* CL2120. Moreover, it was determined from the biosensor experiments based on the SPR effect, the NMR spectroscopy, and molecular modeling that both the Aβ and isoD7-Aβ form a stable non-covalent complex with HAEE, in which one zinc ion is located at the intermolecular interface 11-EVHH-14:HAEE. Previously, using a transgenic mouse model of AD, it was shown that intravenous injections of HAEE highly limit the development of CA in experimental animals [136].

## 5. Conclusions

Taken together, the above data (summarized in Table 1) point at the fundamental role of the non-covalent complexes of zinc ion and isoD7-Aβ as a necessary and sufficient molecular tool that, with the participation of the α4β2 nicotinic acetylcholine receptor, triggers a chain process of pathological aggregation of the endogenous Aβ molecules. The following integral scenario of the molecular mechanism of amyloid plaque formation in transgenic models of AD is suggested (Figure 1). **Phase 1.** Due to unknown reasons (most likely stress and aging), isoD7-Aβ appears in the brain. **Phase 2.** The 11-EVHH-14 region of this chemically modified Aβ isoform interacts with the 35-HAEE-38 region of the α4 subunit of the α4β2-nAChR, resulting in the formation of an “amyloid matrix”, a stable complex on the outer surface of the neuron formed by the isoD7-Aβ and α4β2-nAChR. **Phase 3.** To the “amyloid matrix”, when there is a surge in the concentration of zinc ions in the synaptic cleft, a molecule of the intact Aβ “sticks” according to the zinc-dependent mechanism. The process continues: (i) from the Aβ molecule, the fragment 11-EVHH-14 participates in the interface, and from the “amyloid matrix”, the residues His6 and His13 of the isoD7-Aβ molecule also participate; (ii) an insoluble aggregate appears, on its outer side there is a molecule of endogenous Aβ, which acquires a pathological conformation (due to interaction with the isoD7-Aβ from the “amyloid matrix”); (iii) according to the same scheme, endogenous Aβ molecules newly arriving from the extracellular space “stick” to the Aβ molecules already aggregated on the initial “amyloid matrix”, and the amyloid plaque grows (to a certain canonical volume).

From this mechanism, it follows that the most effective and safe way to destroy such plaques is to utilize structural analogs (peptides or peptidomimetics) of the 35-5HAEE-38 region of the α4 subunit of α4β2-nAChR, e.g., the substance Ac-HAEE-NH_2_, the potential therapeutic effect of which has already been confirmed in the animal AD models [135,136]. When interacting with an amyloid plaque, such analogs will consistently destroy intermolecular interfaces involving the 11-EVHH-14 fragments of Aβ due to their specific binding to the 11-EVHH-14 regions of the aggregated Aβ molecules. In this case, the “amyloid matrix” itself, located at the foundation of this pathological pyramid, will eventually be destroyed, and the neuron, which was burdened with amyloid plaques for many years or even decades, will return to normal life.

## Figures and Tables

**Figure 1 ijms-25-00072-f001:**
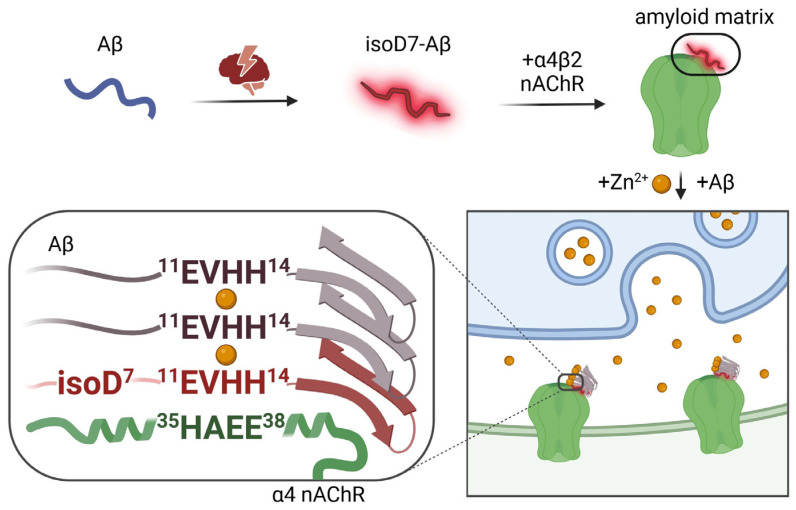
Proposed scenario of the molecular mechanism of amyloid plaque formation in transgenic models of AD.

**Table 1 ijms-25-00072-t001:** Molecular agents involved in the formation of amyloid plaques in transgenic animal models of Alzheimer’s disease.

Agent	Relation to the Formation of Amyloid Plaques	Transgenic Animal Model	References
Human amyloid-beta (Aβ)	The main endogenous building block of the plaques	5xFAD mice	[137]
		APP/PS1 mice	[138]
		Tg2576 mice	[139]
		*Caenorhabditis elegance* CL2120	[140]
Human amyloid-beta with the isomerized Asp7 (isoD7-Aβ)	The most abundant non-enzymatically (due to aging) modified Aβ isoform in the plaques	5xFAD mice	[127,141]
	Endogenous blood component accompanying formation of the plaques	5xFAD mice	[127]
	Exogenous agent accelerating formation of the plaques	APP/PS1 mice	[125]
		5xFAD mice	[126]
α4β2 nicotinic acetylcholine receptor (α4β2-nAChR)	Promoter of the plaques formation	Tg2576 mice	[102]
Zinc ions (Zn^2+^)	Necessary endogenous agent for the plaques formation	Tg2576 mice	[83]
Noncovalent complex between isoD7-Aβ and Zn^2+^	Necessary and sufficient exogenous agent accelerating plaque formation	*Caenorhabditis elegance* CL2120	[135]
Tetrapeptide Acetyl-His-Ala-Glu-Glu-NH_2_ (HAEE)	Exogenous inhibitor of the plaque formation	APP/PS1 mice	[136]
		*Caenorhabditis elegance* CL2120	[135]

## Data Availability

Not applicable.

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
