# Peer review of "Switching On/Off Amyloid Plaque Formation in Transgenic Animal Models of Alzheimer’s Disease"

_ijms, 2023, doi:10.3390/ijms25010072_

Round 1
Reviewer 1 Report
Comments and Suggestions for Authors
In this paper, the authors review mechanisms of extra-cellular aggregation of β-amyloid (Aβ) with a view to preventing this as a neuroprotective strategy in Alzheimer’s disease (AD). The article begins by discussing the pathology of AD and the structure of amyloid plaques which are made up of a nucleus containing Aβ oligomers to which further Aβ molecules becomes attached along with other proteins, nucleic acids and zinc and other metal ions. It then reviews the many available animal models of AD - in particular mouse transgenic models which are most studied – and their advantages and drawbacks. The article then moves on to highlight molecular factors that influence plaque formation in transgenic AD models, in particular Zn2+ ions, α4 subunits of pre-synaptic α4β2 nicotinic receptors, and Aβ with an isomerised aspartate aminoacid (isoD7- Aβ) due to protein aging. Free synaptic Zn2+ ions released by glutamate neuronal firing accelerates Aβ oligomerisation while nicotinic α4 subunits can act as seeds to form amyloid plaque nuclei. IsoD7- Aβ peptide aggregates far faster than Aβ-42 and forms 50% of the amyloid in plaques. It can also act as a nucleation seed. The authors argue that chelating free Zn2+ and blocking α4 subunit access to Aβ with peptides or peptide mimetics will effectively prevent Aβ plaque formation and cite the papers where this has been demonstrated with AD transgenic models.
The paper is well written and instructive. A useful illustration depicting the proposed mechanism of Aβ plaque formation and its prevention is provided. The introduction could be shortened a little as the critical sections start when molecular factors that influence plaque formation are discussed. The authors’ case for preventing extracellular Aβ aggregate formation appears to rest on the successful FDA licensing of lecanemab and now two other monoclonal antibodies directed at amyloid aggregates which all showed effective plaque reduction on PET imaging. However, these agents only partially slowed clinical disease progression. There is now a school of thought (The Amyloid Cascade Hypothesis 2 by V Volloch) that has moved away from regarding the toxicity of extracellular Aβ aggregates as the cause of tau hyperphosphorylation and neuronal death and considers rising intraneuronal Aβ as the main player. The ACH 2 hypothesis explains the limited efficacy to date of drugs targeting extracellular amyloid aggregation. Preventing intracellular Aβ reaching a critical toxic level - possibly by modulating β-secretase activity – is advocated as the way of preventing tau tangle formation and neuronal death. I would be interested to see a discussion about this alternative viewpoint by the current authors.
Author Response
Please, see the attachment

Reviewer 2 Report
Comments and Suggestions for Authors
This is an interesting review article highlighting the role of zinc ions, isoD7-Aβ, and α4β2 nicotinic acetylcholine receptor during the AD-related pathological process (i.e. Aβ aggregation and Aβ seeding). The authors also highlight key molecular mechanisms contributing to Aβ aggregation from studies using transgenic animal AD models. The overall clarity and organization of this review article may need to be improved. Another issue is that authors often use review articles as reference citations in this manuscript. Thus, this review article requires additional revisions to improve organization, accuracy of references, and overall quality.
Other comments:
1. It would be great to include a table to summarize the role of zinc ions, isoD7-Aβ, and α4β2 nicotinic acetylcholine receptor in Aβ aggregation.
Author Response
Please see the attachement
